# Prototype Material for New Strategy of Photon Energy Storage

**Toshio Naito** [1,2,3,4] 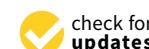

[1]   Graduate School of Science and Engineering, Ehime University, Matsuyama 790-8577, Japan;
tnaito@ehime-u.ac.jp

[2]   Advanced Research Support Center (ADRES), Ehime University, Matsuyama 790-8577, Japan

[3]   Research Unit for Development of Organic Superconductors, Ehime University, Matsuyama 790-8577, Japan

[4]   Geodynamics Research Center, Ehime University, Matsuyama 790-8577, Japan

**Abstract:** The smart utilization of photons is paid global attention from the viewpoint of renewable energy and information technology. However, it is still impossible to store photons as batteries and condensers do for electrons. All the present technologies utilize (the energy of) photons in situ, such as solar panels, or in spontaneous relaxation processes, such as photoluminescence. If we can store the energy of photons over an arbitrary period and utilize them on demand, not only we will make an innovative progress in energy management, but we will also be able to replace a part of electrons by photons in the information technology for more efficient performance. In this article, we review a prototype of such a material including the current status of related research as well as where we are heading for.

**Keywords:** phase transitions under UV-irradiation; photoexcited states; charge-transfer complexes; metal-dithiolene complexes; non-planar Au(III)-complexes; reversible photoreactivity; Au(III)-complex radical anions; bipyridyl derivatives

---

## 1. Introduction

### 1.1. Photoluminescence and Related Phenomena

Photoexcited states and subsequent relaxation processes, which are generally quick and complicated series, often produce otherwise impossible unique and interesting phenomena. Examples include transient metastable states and physical properties such as various types of photoluminescence [1]. In photo-induced and photoexcited phenomena, such as photoconductivity, some of them proceed spontaneously under irradiation, while others are driven by external fields in addition to light. In most cases, the incident photon energy is immediately transformed into work and heat in the irradiated materials and is consumed on the spot. Some photoexited states and properties can be maintained as long as the irradiation is continued, while some of them retain the photoexcited properties after cessation of irradiation [2,3]. Meanwhile, in this review, we will focus on the photoexcited states and properties that persist even after irradiation is ceased. Of particular interest are the materials that can retain the photoexcited states without spontaneous relaxation and can release the energy on demand. Such a material is unknown, yet if it exists, it enables us to use photons as we use electrons in a battery. Such a technique is tentatively named "photon energy storage (PES)" in this review. Based on PES, not only we will make innovative progress in energy management, but we will also be able to replace a part of electrons by photons in the information technology for more efficient performance.

First, we shall briefly review recent studies on the photoluminescence and related phenomena to distinguish the PES from existing phenomena and techniques. Common relaxation processes after

irradiation of various wavelengths on various materials involve non-radiative decay processes, where the excitation energies received from the photons are completely transformed into heat and dissipated to surroundings by phonons, i.e., thermal vibration of atoms and molecules. However, some materials spontaneously emit light under irradiation of proper wavelengths of light (generally UV or even higher energy light), which is generally called photoluminescence. The well-known phenomena of photoluminescence are fluorescence and phosphorescence (Figure 1) [4]. Both include radiative decay processes, during which molecules give off photons after receiving the excitation energies under irradiation. The empirical difference between them lies in the duration of decay, i.e., the relaxation time. Usually, the light is emitted within a few nanoseconds after cessation of irradiation in both types of photoluminescence. The emission rapidly decreases in intensity in fluorescence, while it slowly decreases in phosphorescence enabling weak but long emission (typically a few seconds, sometimes a few hours). The duration of emission depends on the temperature. Hereafter, the relaxation times of photoluminescence are those at room temperature (RT) unless otherwise noted. The differences in the relaxation times originate from partial relaxation by non-radiative decay in the initial stage of phosphorescence, during which the molecules change their geometries, i.e., molecular distortion occurs. If an intersystem crossing (ISC), i.e., non-radiative decay between two states with different spin multiplicities such as $S_1$ to $T_1$, can take place in this process, the subsequent photon emission becomes extremely slow as stated above (from a few seconds to a few hours), as optical transitions (radiative decay processes) between the two states with different spin multiplicities ($T_1 \rightarrow S_0$) are forbidden. In fluorescence, the relaxation for emission occurs between two states with the same spin multiplicities such as $S_1 \rightarrow S_0$, which is allowed and thus occurs immediately after the photoexcitation, enabling intense but transient emission (typically, from nanoseconds to sub-nanoseconds). Accordingly, the initial non-radiative decay process(es) in phosphorescence give(s) us the time during which the received photon energy is stored in the material leaking a part of the energy.

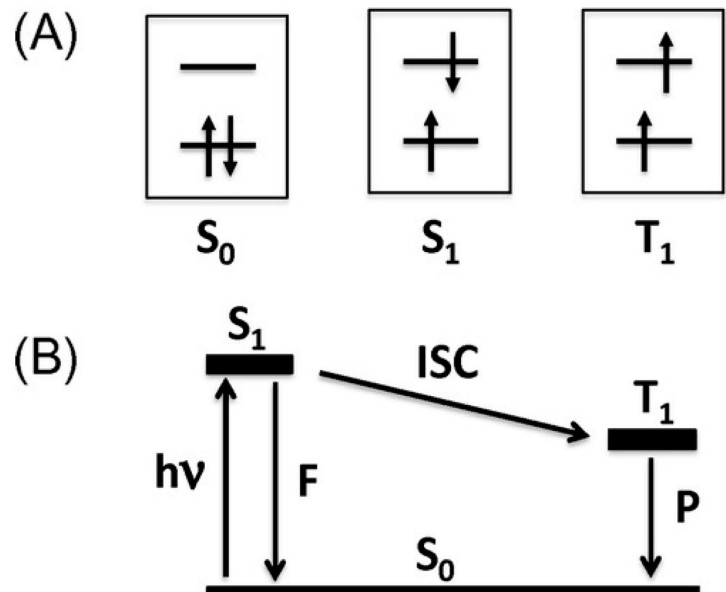

**Figure 1.** (**A**) Electron configurations of a singlet ground state ($S_0$), a singlet excited state ($S_1$), and a triplet excited state ($T_1$). (**B**) Jablonski diagram for fluorescence (F), intersystem crossing (ISC), and phosphorescence (P) emissions. hν designates photoexcitation. Reproduced from Ref. [4] with permission. Copyright 2018, The Chemical Society of Japan.

From the viewpoint of PES, a phenomenon called persistent luminescence (PersL) [5] has been paid particular attention over the last two decades, beginning with the discovery of $SrAl_2O_4:Eu^{2+}$-$Dy^{3+}$ [6] and related inorganic compounds. PersL was originally called "afterglow", already known in the Song dynasty in ancient China, dating back a thousand years. Now there are many kinds of inorganic PersL

materials, called inorganic phosphors, known and applied in practical use [5,7]. Still, the mechanism is under debate. They slowly release stored energy as light during a few minutes to several days with a possibly different mechanism from that of phosphorescence, as the temperature-dependence of relaxation time and the emission spectra are qualitatively different between PersL and phosphorescence even for the same material capable of both types of luminescence under the same wavelength of photoexcitation [8]. In PersL, instead of the ISC in phosphorescence, a trap state plays an indispensable role for storing the energy and delaying the light emission (Figure 2), where the trap state is considered to be provided by lattice defects in most cases [5].

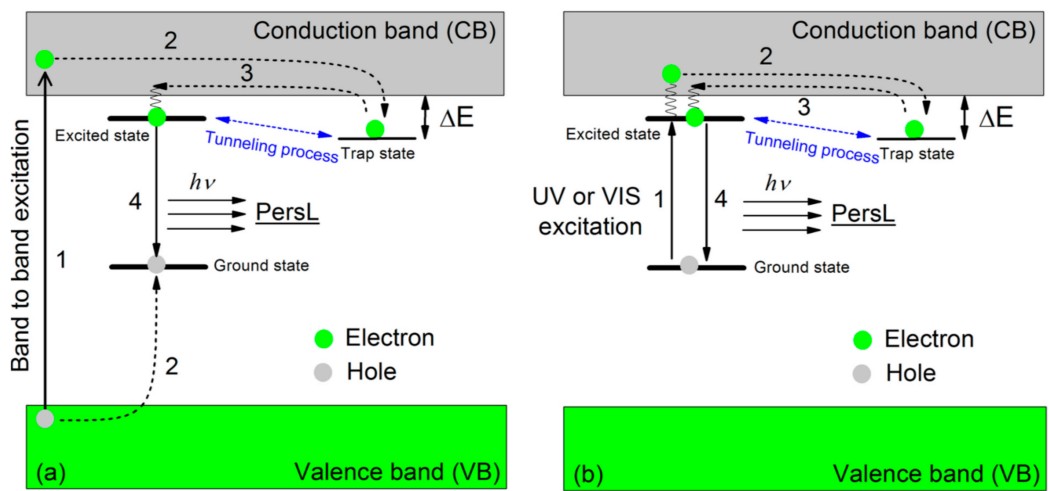

**Figure 2.** Schematic illustration showing the proposed mechanisms of PersL in inorganic phosphors (electron trapping-detrapping model): (**a**) Band-to-band excitation by high energy light; (**b**) UV or visible light excitation. Reproduced from Ref. [5] with permission. Copyright 2018, Elsevier.

Because of such a long storage time, PersL phosphors are sometimes referred to as "optical batteries" [9,10].

Similar to PersL, some organic charge-transfer (CT) complexes are also known to exhibit long-lived luminescence called "long-persistent luminescence" (LPL) [11]. A proposed mechanism of organic LPL (OLPL) is clearly different from that of phosphorescence (Figure 3). In fact, the time-dependence of emission intensity is qualitatively different from that of phosphorescence. In OLPL, the materials slowly release stored exciton energy as light, having a thousand times longer life of emission (over 1 h in some cases) after cessation of photoexcitation. In this slow emission, an intermediate state plays an indispensable role. Here the intermediate states contain the lowest singlet ($^1$CT) and triplet ($^3$CT) excited-states of the photo-generated exciplex as well as a partially charge-separated state (CSS) formed by a pair of a radical cation and anion dissociated from the exciplex [11]. They serve as trapped states similar to defects in PersL. As a closely related topic to OLPL, thermally activated delayed fluorescence (TADF) is also extensively examined [12]. However, it is associated with the organic light-emitting diodes (OLEDs) rather than the phosphorescent materials; the OLEDs convert electricity to light. Thus, we shall not go into details here.

All of the aforementioned examples are closely related to PES in that they are pursuing the materials that emit light as long as possible after cessation of external photoexcitation. However, they are clearly different from PES; PES requires the control of emission of light, instead of spontaneous emission. As the ultimate goal of PES, it is desirable that one can commence, interrupt, resume, and adjust the light intensity of emission on demand by some triggers. At present, the only candidate material of PES discussed below, BPY[Au(dmit)$_2$]$_2$ (**1**; BPY$^{2+}$ = *N*,*N*′-ethylene 2,2′-bipyridinium, dmit$^{2-}$ = 2-thioxo-1,3-dithiole-4,5-dithiolato) [13], does not emit light after photoexcitation at RT. Thus, we could not discuss the luminescence property of **1** here. Instead, the possibility of PES lying in **1** is discussed based on its structural and physical properties.

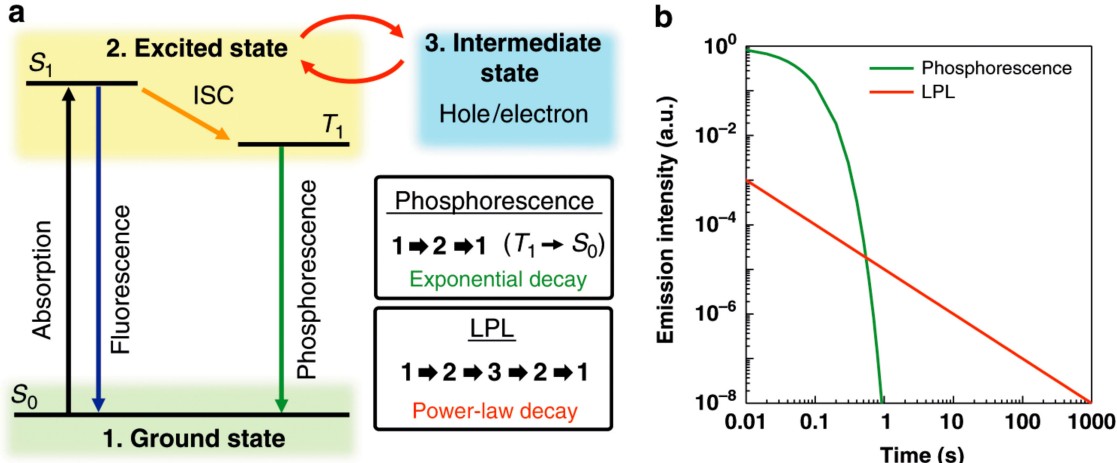

**Figure 3.** Differences between long-persistent luminescence (LPL) and phosphorescence: (**a**) Schematic diagram of fluorescence, phosphorescence, and LPL; (**b**) Ideal emission decay profiles of phosphorescence and LPL on logarithmic plots. Reproduced from Ref. [11], which is licensed under the Creative Commons Attribution 4.0 International License.

### 1.2. Au(III)-Dithiolene Complexes

It is well known that the transition metal-complexes of heavy-metal ions with $d^6$, $d^8$, and $d^{10}$ electron configurations such as Ir(III), Pt(II), and Au(I) are advantageous for designing phosphorescent materials (heavy atom effect) [14]. Although the number of papers on the phosphorescence concerning the square-planar Au(III) complexes is much smaller than that of the aforementioned complexes, they have also been studied for photoluminescent materials [15]. From various pharmaceutical and medical points of view, the synthesis, molecular structures, and reactivity have been extensively studied on the square-planar Au(III) complexes [16–18]. Various Au(III)-dithiolene complexes have been synthesized and characterized for conducting materials [19,20]. Owing to these efforts in independent fields, the Au(III) complexes are one of the most studied metal-complexes in terms of structural, chemical, and physical properties. Here, considering the relevance to the discussion below, we shall basically limit ourselves to the structural and physical properties of the CT salts of Au(III)-dithiolene complexes ($[AuL_2]^{n-}$; $n$ = 0–1) to understand their general features to be compared with those of **1** in the subsequent subsections. Figure 4 shows such ligands and their square-planar Au(III)-complex molecules selected mainly from recent work for molecular conductors and related functional materials. For other metal-dithiolene complexes, readers are asked to consult with other reviews [19,21–25]. Although the terminology "CT salts" is now less common than "CT complexes", the former is used below for clarity; the "complex(es)" designates the metal-complex molecule(s), while "salt(s)" designates the solid state(s) produced by (partial) redox reactions between metal complexes and other chemical species.

## Donors and Cations

ET　　　　　　TMTSF　　　　　　TTF　　　　　　$[Fe(cp)_2]^{x+}$

MV$^{2+}$　　　　　　BPY$^{2+}$　　　　　　NMP$^+$

## Ligands (L) and Au-Complexes (AuL$_2$)

mnt$^{2-}$　　dmit$^{2-}$　　dddt$^{2-}$　　tpdt$^{2-}$　　F$_2$pdt$^{2-}$

R = CH$_3$ ; dmdt$^{2-}$
R = -(CH$_2$)$_3$- ; tmdt$^{2-}$
R = -S(CH$_2$)$_3$S- ; ptdt$^{2-}$
R = CF$_3$ ; hfdt$^{2-}$

HC=CH ; α-tdt$^{2-}$
H$_2$C-CH$_2$ ; dtdt$^{2-}$

bdt$^{2-}$　　ttdt$^{2-}$

α-mtdt$^{2-}$　　Au(α-tpdt)$_2$　　Au(*S*,*S*-dm-dddt)$_2$

R-tzdt$^{2-}$

R = I, C$_2$H$_5$ (Et),
　S(CH$_2$)$_2$OH (HOEtS)

Au(dtpdt)$_2$　　Au(*R*,*R*-dm-dddt)$_2$

Au(R-thiazdt)$_2$

R = CH$_3$ (Me), C$_2$H$_5$ (Et),
　CH(CH$_3$)$_2$ (*i*Pr),
　CH(CH$_2$)$_2$ (cPr)
　C$_3$H$_7$ (*n*Pr)
　C$_2$H$_4$OH (EtOH)
　N(CH$_3$)$_2$ (NMe$_2$)

R = CH$_3$ ; Au(α-Me$_2$tpdt)$_2$
R = C$_2$H$_5$ ; Au(α-EtMetpdt)$_2$

(X, Y) = (S, S) ; S$_4$(=S)$_2$
　　(Et-thiazdt)
(X, Y) = (S, O) ; S$_4$(=O)$_2$
　　(Et-thiazodt)
(X, Y) = (Se, S) ; Se$_4$(=S)$_2$
(X, Y) = (Se, O) ; Se$_4$(=O)$_2$

**Figure 4.** Chemical structures of selected dithiolene ligands (L) and their gold complexes (AuL$_2$) discussed in this paper. Unless there are isomers for AuL$_2$, only L is shown; otherwise, only the relevant isomers to those in Tables 1 and 2 are shown. There are some ligands without particular abbreviations.

The [Au(III)L$_2$]$^-$ salts were known since 1963 [26], which was six years since the synthesis of the sodium salts of some kinds of ethylenedithiolate (maleonitrile (mnt) and its derivatives; Figure 4) were reported [27], one year since the chemistry of the free ligand (Na$_2$mnt) was extensively

examined [28], and also one year since the structures, electronic configurations, and magnetic properties of a series of the transition metals (Ni(II), Pd(II), Pt(II), Cu(II), Zn(II), and Co(II)) mnt complexes were reported [29]. Then, the advent of dithiolene ligands with more extended $\pi$-conjugation, beginning with dimercaptoisotrithione (dmit) [30,31], paved the avenue to study the solid state chemistry and physics of molecular conductors and related materials based on $ML_2$ (Tables 1 and 2). The first series of intensive study on the salts containing $[Au(III)L_2]^{n-}$ ($0 < n \leq 1$) dates back to ~1990 as new promising building blocks for conducting materials [32–39] by analogy with the CT salts of the same $d^8$ square-planar complexes $[M(II)L_2]^{m-}$ (M = Ni, Pd, and Pt; $m = 0$–2), since the latter group of compounds successively produced a number of superconductors in 1985–1993 [21,22]. The preliminary measurements demonstrated that many $Au(III)L_2$ salts were unusually highly conducting with metallic temperature-dependence [37] and that some of them remained to be metallic down to 4 K or even lower temperatures [36]. However, since the stability during (partial) oxidation in solution is quite different between $[Au(III)L_2]^{n-}$ and $[M(II)L_2]^{m-}$ [34,40], the single crystals of the $[AuL_2]^{n-}$ salts with well-defined stoichiometry and good quality for X-ray structural analysis were hardly obtained then. Consequently, the basic information such as crystal structures and chemical compositions of many of the $[AuL_2]^{n-}$ salts remained unknown, preventing the researchers then from further and detailed examination of the solid state properties. Among $Au(III)L_2$ complexes, there are unique exceptions in that cation radical salts of $[Au(III)L_2]^{k+}$ ($0 < k$) are isolated when L = 5,6-dihydro-1,4-dithiin-2,3-dithiolate (dddt; Figure 4) [41]. Since the molecular structure is similar to and the formal charge is the same with $ET^{0.5+}$ (ET = bis(ethylenedithio)-tetrathiafulvalene; Figure 4), the $[M(dddt)_2]^{0.5+}$ (M = Ni, Pd, Pt) salts exhibit similar crystal structures with the metallic salts of $ET_2X$ (X = various monoanions) [42] and high conductivities in fact. However, because of the inferior crystal qualities, the crystal structures and chemical compositions remain unknown for the $Au(dddt)_2$ salts. Consequently, most of the papers on the cationic $M(dddt)_2$ salts concern M = Ni, Pd, and Pt, and the solid state properties of $Au(dddt)_2$ salts [41] and neutral $Au(dddt)_2$ (Figure 5) [43] are scarcely known. There are also extensive studies on unsymmetrical Au(III)-dithiolene ($[Au(III)L_1L_2]^n$; $n = -1 \sim 0$) and chiral Au(III)-dithiolene complexes reported to date [19,44,45]. However, we shall not discuss them here, since they are less relevant to the following discussion.

**Table 1.** Crystalline Gold(III)-dithiolene complexes containing ionic $[Au(III)L_2]^n$ ($-1 \leq n \leq 1$, $n \neq 0$) [1].

| L [2] | Formal Charge ($n$) | Counter Ion [3] | Stoichiometry (Cation: Anion) | Structural Feature [4] | Solid State Properties [5] | References |
|---|---|---|---|---|---|---|
| mnt | −1 | $Et_4N^+$ | 1:1 | - | Dia | [26] |
| mnt | −1 | $NMP^+$ | 1:1 | Mixed stacking | - | [46] |
| dmit | −1 | $^nBu_4N^+$ | 1:1 | Related to **1** | L [6] | [34] |
| dmit | −0.22 | $^nBu_4N^+$ | 0.22:1 | - | Sem | [34] |
| dmit | - | TTF | 1:1 | - | Sem | [34] |
| dmit | - | $Fe(cp)_2$ | 0.25:1 | - | Sem | [34] |
| dmit | - | TTF | 3:2 | - | Sem * | [35,47] |
| dmit | - | TMTSF | 3:2 | - | Sem * | [35,47] |
| dmit | - | ET | 1:1 | - | Sem * | [35,47] |
| dmit | −0.5 | $Et_4N^+$ | 1:2 (α) | - | Met | [36] |
| dmit | −0.5 | $Et_4N^+$ | 1:2 (β) | 2D | Sem | [36] |
| dmit | −0.5 | $^nBu_4N^+$ | 1:2 (α) | - | H [6] | [36] |
| dmit | - | $^nBu_4N^+$ | unknown (β) | - | Met | [36] |
| dmit | - | $^nBuEt_3N^+$ | - | - | Met (~220 K) | [36] |
| dmit | - | $^nBu_2Et_2N^+$ | - | - | Met (~120 K) | [36] |
| dmit | - | $^nBu_3EtN^+$ | - | - | Sem | [36] |
| dmit | −0.5 | $Me_4N^+$ | 1:2 | - | Sem | [36] |
| dmit | - | $Me_3S^+$ | unknown (α) | - | Sem | [36] |
| dmit | - | $Me_3S^+$ | unknown (β) | - | Met (~90 K) | [36] |
| dmit | - | $Cat^+$ | - | - | Met (~100 K) | [36] |
| dmit | - | $^nBu_3S^+$ | - | - | Met | [36] |
| dmit | −0.5 | $Li^+$ | 1:2 | - | Met (~160 K) | [37] |
| dmit | −0.5 | $Na^+$ | 1:2 | - | Met (~83 K) | [37] |
| dmit | −0.5 | $K^+$ | 1:2 | - | Met (~60 K) | [37] |
| dmit | −0.5 | $Me_4N^+$ | 1:2 | Disordered anions | Sem | [38] |
| dddt | - | $BF_4^-$ | - | - | H [6] | [41] |
| dddt | - | $IBr_2^-$ | - | - | H [6] | [41] |
| dddt | - | $SnCl_6^{x-}$ | - | - | Met (~150 K) | [41] |
| dddt | - | TTF | 1:1 | Disordered ethylene-group in anions | - | [48] |
| Me-thiazdt | −1 | $Et_4N^+$ | 1:1 | Disordered anions | - | [20] |

**Table 1.** *Cont.*

| L [2] | Formal Charge (*n*) | Counter Ion [3] | Stoichiometry (Cation: Anion) | Structural Feature [4] | Solid State Properties [5] | References |
|---|---|---|---|---|---|---|
| Et-thiazdt [7] | −1 | $Et_4N^+$ | 1:1 | - | - | [49] |
| *n*Pr-thiazdt | −1 | $Ph_4P^+$ | 1:1 | *cis/trans* [8] | - | [50] |
| *i*Pr-thiazdt | −1 | $Ph_4P^+$ | 1:1 | *trans* | - | [51] |
| *c*Pr-thiazdt | −1 | $Et_4N^+$ | 1:1 | *trans* | - | [50] |
| EtOH-thiazdt | −1 | $Et_4N^+$ | 1:1 | *cis/trans* | - | [52] |
| $NMe_2$-thiazdt | −1 | $Ph_4P^+$ | 1:1 | *trans* | - | [50] |
| I-tzdt | −1 | $^nBu_4N^+$ | 1:1 | *cis/trans* [9] | Dia | [53] |
| HOEtS-tzdt | −1 | $Ph_4P^+$ | 1:1 | *trans* | - | [54] |
| dm-dddt | −1 | $^nBu_4N^+$ | 1:1 | Enantiopure separated anions | - | [44] |
| α-EtMetpdt | −1 | $Ph_4P^+$ | 1:1 | *trans* | - | [55] |
| α-$Me_2$tpdt | −1 | $Ph_4P^+$ | 1:1 | Disordered anions [10] | - | [55] |
| bdt | −1 | $^nBu_4N^+$ | 1:1 | - | - | [56] |
| $S_4(=O)_2$ [11] | −1 | $Ph_4P^+$ | 1:1 | Separated anions | - | [57] |
| $Se_4(=S)_2$ | −1 | $Et_4N^+$ | 1:1 | Separated anions | - | [57] |
| $Se_4(=O)_2$ | −1 | $Ph_4P^+$ [12] | 1:1 | Separated anions | - | [57] |
| dtpdt | −1 | $^nBu_4N^+$ | 1:1 | Zigzag chains | Dia | [40] |
| α-tpdt | −1 | $^nBu_4N^+$ | 1:1 | - | Dia | [40] |
| tpdt | −1 | $^nBu_4N^+$ | 1:1 | 2D | Dia | [40] |
| tpdt | ~−0.5 | $^nBu_4N^+$ | ~1:2 | - | Curie | [40] |
| $F_2$pdt | −1 | $^nBu_4N^+$ | 1:1 | Separated anions | - | [58] |
| α-mtdt | −1 | $^nBu_4N^+$ | 1:1 | *cis-trans* disorder | Dia | [59] |
| α-tbtdt | −2 | $^nBu_4N^+$ | 2:1 | Dinuclear Au(I) | - | [59] |
| ttdt | −1 | $^nBu_4N^+$ | 1:1 | - | - | [60] |
| ttdt | - | TTF | 1:1 | Separated dimers of cations and anions | - | [60] |

[1] Basically the table does not include intermediates in the syntheses of neutral gold complexes, if the solid-state properties of the former are not characterized. [2] Abbreviated names. They are sometimes designated by capital letters. For chemical structures, see Figure 4. [3] $Ph = C_6H_5$, $^nBu = {}^nC_4H_9$, $Et = C_2H_5$, $Me = CH_3$, $Cat^+ = Me_3NC_3H_5^+$, [4] Keywords for describing the structural features (molecular structures or arrangements of the $AuL_2$ species) when such details are available in the reference(s). 1D = one-dimensional, 2D = two-dimensional. [5] Magnetic and/or conducting properties are indicated. Dia = diamagnetic solid, Ins = insulator, Sem = semiconductor, Met = metal with the metal-nonmetal transition temperature in parentheses if it exhibits such a transition. Properties based on compaction pellets are shown with asterisks. [6] Only the room temperature properties are available, and the temperature-dependences remain to be clarified. H = highly conducting, L = low conducting. [7] Furthermore, it is named as $S_4(=S)_2$ in other references. [8] With regard to the orientation of the nitrogen substituents, the Au complex assumes *trans*-conformation when crystal solvent ($CH_3CN$) is incorporated, while *cis*-conformation without $CH_3CN$. [9] The Au-complex assumes *trans*-conformation when it is recrystallized from acetone, while *cis*-conformation from $CH_3CN$. [10] Disorder is found only in one of two crystallographically independent anions. [11] Furthermore, it is named as Et-thiazodt in other references. [12] Although the cation is designated $Ph_4P^+$ in the main text, the actual cation in the deposited cif file is $Et_4N^+$.

**Table 2.** Crystalline Gold(III)-dithiolene complexes containing neutral [Au(III)$L_2$].

| L [1] | Structural Feature [2] | Conducting Properties [3] | Magnetic Properties [4] | References |
|---|---|---|---|---|
| dddt | Figure 5a | L | Dia | [43] |
| hfdt | Dimerized 2D | Sem | Almost Dia | [61] |
| tmdt | 3D | H * | AF (100–110 K) | [62–64] |
| dmdt | - | Met (~50 K) * | Pauli (~50 K) | [64] |
| ptdt | 2D | Met * | Pauli (~80 K) | [65] |
| Me-thiazdt | 3D | Met | Pauli | [20] |
| Et-thiazdt [5] | 2D | Sem | Pauli | [49,66] |
| *n*Pr-thiazdt | *trans*, Uniform 1D | Sem | - | [50] |
| *c*Pr-thiazdt | *trans*, Uniform 1D | Sem | - | [50] |
| *i*Pr-thiazdt | Crisscross structure | Sem | Pauli | [51] |
| EtOH-thiazdt | 2D | Sem | CW | [52] |
| $NMe_2$-thiazdt | *trans*, Crisscross structure | Sem | - | [50] |
| HOEtS-tzdt | Uniform 1D | Mott Ins | - | [54] |
| dm-dddt | Asymmetry in ligands | Sem | - | [44] |
| $OC_4$ | Dimerized 1D | Sem | ST | [67] |
| α-EtMetpdt | Uniform 1D | Sem | - | [55] |
| α-$Me_2$tpdt | - | Sem * | - | [55] |
| bdt | 1D with superstructure | Sem | Complicated behavior | [56,68] |
| $S_4(=O)_2$ [6] | 1D | Sem | BF | [57] |
| $Se_4(=S)_2$ | Q2D | Sem | Pauli | [57] |
| $Se_4(=O)_2$ | Q2D | Sem | Pauli | [57] |
| $F_2$pdt | Uniform 1D | Sem | BF | [58] |
| α-tpdt | - | Met (~15 K) * | Pauli | [40] |
| dtpdt | - | L * | Curie | [40] |

**Table 2.** *Cont.*

| L [1] | Structural Feature [2] | Conducting Properties [3] | Magnetic Properties [4] | References |
|---|---|---|---|---|
| α-tdt | - | Almost Met * | Large $\chi$ [7] | [59,69] |
| α-mtdt | - | Sem * | Pauli | [59] |
| dtdt | - | Almost Met * | Large $\chi$ [7] | [59,69] |

[1] Abbreviated names. They are sometimes designated by capital letters. For chemical structures, see Figure 4. $i$Pr = CH(CH$_3$)$_2$, Et = C$_2$H$_5$, Me = CH$_3$. [2] Keywords for describing the structural features (molecular structures or arrangements of the AuL$_2$ species) when such details are available in the reference(s). 1D = one-dimensional, 2D = two-dimensional, Q2D = quasi-two-dimensional, 3D = three-dimensional. [3] Conducting properties under ambient pressure. Ins = insulator, Mott Ins = Mott insulator, Sem = semiconductor, Met = metal with the metal-nonmetal transition temperature in parentheses if it exhibits such a transition. H = highly conducting without mentioning whether it is a semiconductor or a metal, L = low conducting even at room temperature. Properties based on compaction pellets are shown with asterisks. [4] Dia = diamagnetic solid, AF = antiferromagnets with Néel temperatures in parentheses, Pauli = Pauli or similar temperature-independent paramagnetism, CW = Curie-Weiss behaviour, ST = singlet-triplet behaviour, BF = Bonner-Fischer behaviour, Curie behaviour. Although not referred in this table, Curie behavior at lowest temperature is observed in most of these compounds. [5] Furthermore, it is named as S$_4$(=S)$_2$ in other references. [6] Furthermore, it is named as Et-thiazodt in other references. [7] Temperature-dependent large susceptibility was observed, which remains to be explained.

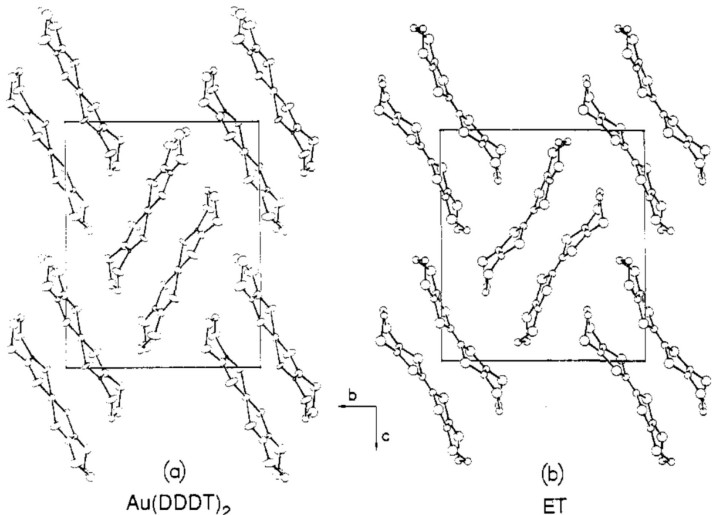

**Figure 5.** Crystal structure (*bc*-plane) of neutral Au(DDDT)$_2$ compared with that of neutral ET: (**a**) Au(DDDT)$_2$; (**b**) ET (reported originally in Ref. [70]). Reproduced from Ref. [43] with permission. Copyright 1987, American Chemical Society.

About 10–20 years have passed since then, when many researchers resumed intensive studies on the conducting and magnetic properties of the Au(III)L$_2$ salts. More exactly, instead of the salts of the anions [AuL$_2$]$^{x-}$ (0 < $x$ ≤ 1), the neutral radical species [AuL$_2$]$^0$ have been paid particular attention for the last ~5–10 years. From the chemical point of view, such a research trend partly originates from the redox property of [Au(III)L$_2$]$^-$, which is different from other [M(II)L$_2$]$^{m-}$ (M = Ni, Pd, and Pt; $m$ = 0–2) [34,40]. While the radical anions with non-integer charges, [ML$_2$]$^{n-}$ (0 < $n$ < 1), appear to be more stable in the case of M = Ni, Pd, and Pt than M = Au, the [AuL$_2$]$^-$, anions are sometimes readily to produce stable neutral [AuL$_2$]$^0$ species during (electro)chemical oxidation [34,40]. This feature is highly advantageous for the development of the single-component molecular (SCM) conductors, where a single kind of neutral radical species [ML$_2$]$^0$ (M = Ni, Cu, Au; for L see Figure 4) comprises a (semi-, metallic or super-)conductor (for [AuL$_2$]$^0$, see Table 2) [19,20,23–25,40,44,49,51,52,55,57,58,61–68,71,72]. In fact, there are a number of reports on newly synthesized SCM metals and semiconductors based on neutral radical AuL$_2$ with various kinds of L. Their single crystals, having well-defined chemical compositions and being stable under normal atmosphere at RT, are generally of sufficient dimensions and qualities for X-ray structural analyses and physical measurements of solid-state properties. Thus, they are well characterized regarding their structural, electrical, magnetic, and optical properties. Typical crystal structures are shown in Figures 6–8. They share the structural features that the component molecules

$[AuL_2]^0$ are all planar (except for alkyl substituents if they have) and of elongated geometries, and that they aggregate themselves into the closest packing structures with π–π overlaps. However, as is always the case with the molecular crystals, the resultant crystal structures are rather different from each other reflecting the details of the structures.

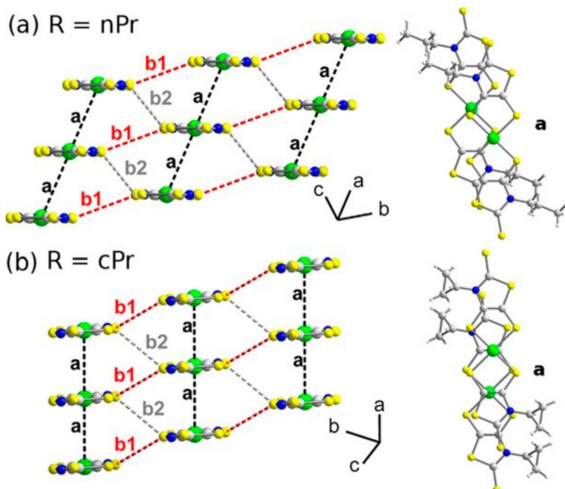

**Figure 6.** One-dimensional molecular arrangement in the crystals of Au(R-thiazdt)₂: (**a**) R = *n*Pr; (**b**) R = cPr. For R-thiazdt, see Figure 4. The dotted lines show intermolecular interactions designated as a, b1 and b2. Compared with a, b1 and b2 are evidently small, resulting in a columnar structure along the *a*-axis in both compounds. Reproduced from Ref. [50] with permission. Copyright 2017, Wiley.

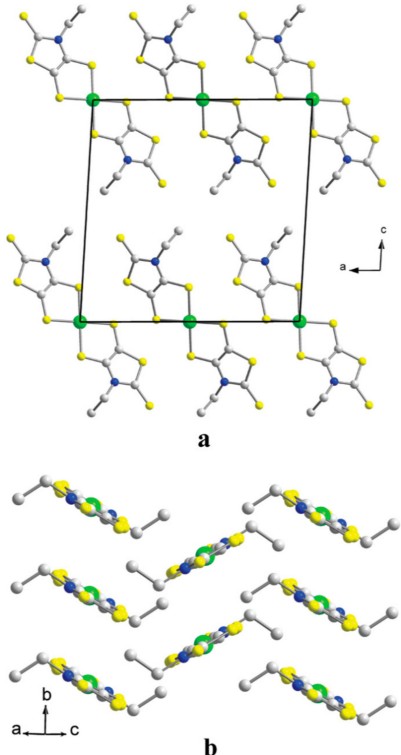

**Figure 7.** Two-dimensional molecular arrangement in the crystals of Au(Et-thiazdt)₂: (**a**) view along the *b*-axis; (**b**) view along the 101-direction. For Et-thiazdt, see Figure 4. The molecules interact with each other in an isotropic way in the *ab*-plane, resulting in a sheet structure in the *ab*-plane. Reproduced from Ref. [49] with permission. Copyright 2009, American Chemical Society.

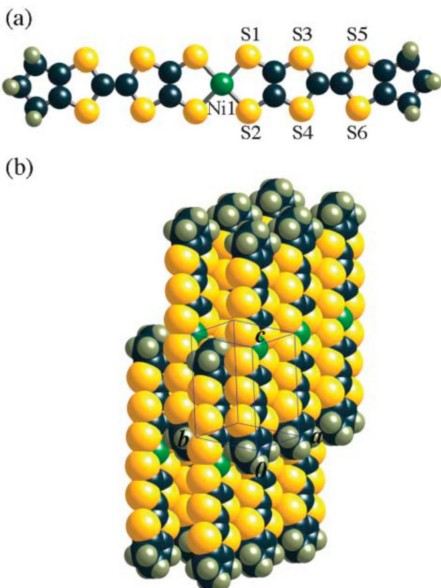

**Figure 8.** Three-dimensional molecular arrangement in the crystals of Ni(tmdt)$_2$: (**a**) molecular structure; (**b**) crystal structure. For tmdt, see Figure 4. The closely packed molecules interact with each other in a three-dimensional way. Reproduced from Ref. [62] with permission from the Royal Society of Chemistry. Copyright 2005.

## 2. Physical Properties of BPY[Au(dmit)$_2$]$_2$ (1): An Overview

### 2.1. Crystal and Molecular Structure

The crystal structure of **1** is shown in Figure 9 [13]. The asymmetric unit comprises two [Au(dmit)$_2$]$^{\delta-}$ anions and one BPY$^{2\delta+}$ cation (Figure 10) [13]. For simplicity, the formal charge on the BPY cation will be denoted as 2+ below, and similarly [Au(dmit)$_2$]$^{\delta-}$ as [Au(dmit)$_2$]$^-$, unless this designation is misleading. The short molecular axes of two crystallographically independent Au(dmit)$_2$ anions are arranged almost perpendicularly to each other (Figures 10 and 11) [13]. Such orthogonal arrangement of metal(M)-dithiolene(L) complex molecules (M = Zn, Ni, Pd, Pt, Cu, Au, etc) along their short molecular axes has been scarcely observed in a great number of related complexes reported thus far [19–25,38,40–46,48–77] except for $^n$Bu$_4$N[Au(dmit)$_2$] [34]. One of the Au(dmit)$_2$ anions faces a BPY cation, and they together form mixed-stacking columns. Only the Au(dmit)$_2$ anions in the mixed-stacking columns contain Au atoms (Au2) that adopt both planar (P, Au2A) and non-planar (NP, Au2B) geometries in a disordered manner (Figure 10) [13]. The remaining Au atoms (Au1) are not disordered and adopt the P geometry. The Au(III) ions with d$^8$ configuration have been generally known to take square-planar coordinates, and non-planar geometry is highly exceptional. The NP geometry and the equilibrium between P and NP geometries at the Au2 site distinguish **1** from all other known compounds whether organic or inorganic. There are many short contacts between sulfur atoms as well as between gold and carbon atoms. Thus, there could be interaction between anions in addition to that between anions and cations. Such a network is favorable for properly isolating and stabilizing the NP [Au(dmit)$_2$]$^{\delta-}$ species at the same time, which is indispensable for PES. The distances from Au2B to the nearest C atoms on BPY (3.037–3.46 Å; Au2B–C17 and Au2B–C18; for atomic numbering scheme, see Figure 12) are comparable or shorter than the sum of the corresponding van der Waals radii (Au–C; d$_{vdW}$(Au–C) = 3.36 Å) at all temperatures examined (Figure 12) [13]. In particular, one of the Au–C distances (Au2B–C18), which is greater than d$_{vdW}$(Au–C) above 280 K, rapidly decreases to be less than d$_{vdW}$(Au–C) below ~280 K. Although the mixed stacking structures comprised of $\pi$-conjugated anions and cations are commonly observed, shorter distances than van der Waals distances between metal centers and carbon atoms are unusual.

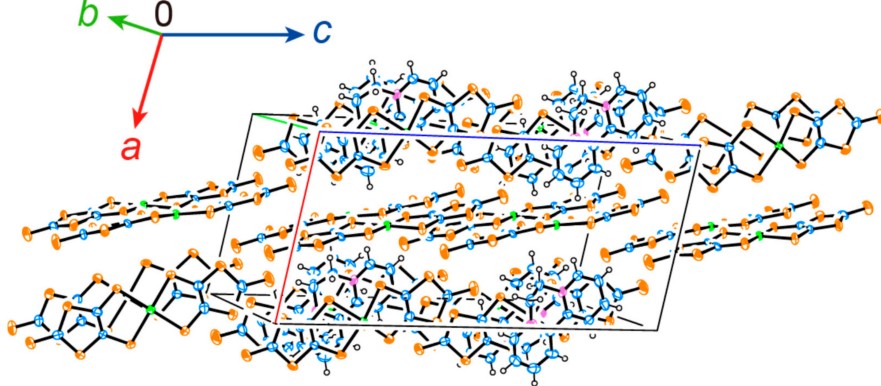

**Figure 9.** Crystal structure of BPY[Au(dmit)$_2$]$_2$ [13]. The orange, blue, violet, and green spheres designate sulfur, carbon, nitrogen, and gold atoms, respectively. Hydrogen atoms are designated by black open circles.

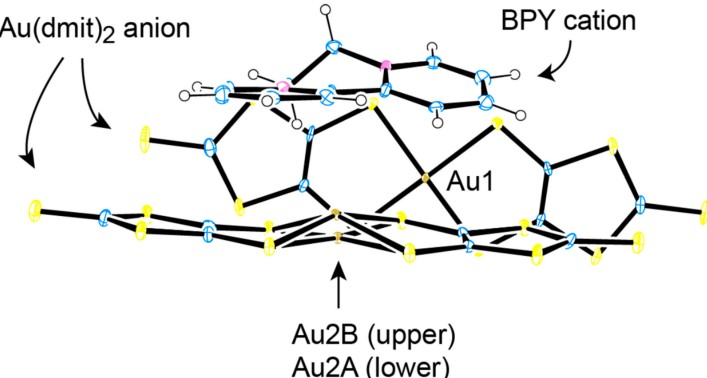

**Figure 10.** Crystallographically independent two Au(dmit)$_2$ anions and the BPY cation in BPY[Au(dmit)$_2$]$_2$ [13]. The yellow, blue, and light brown spheres designate sulfur, carbon, and gold atoms, respectively. Hydrogen atoms are designated black open circles.

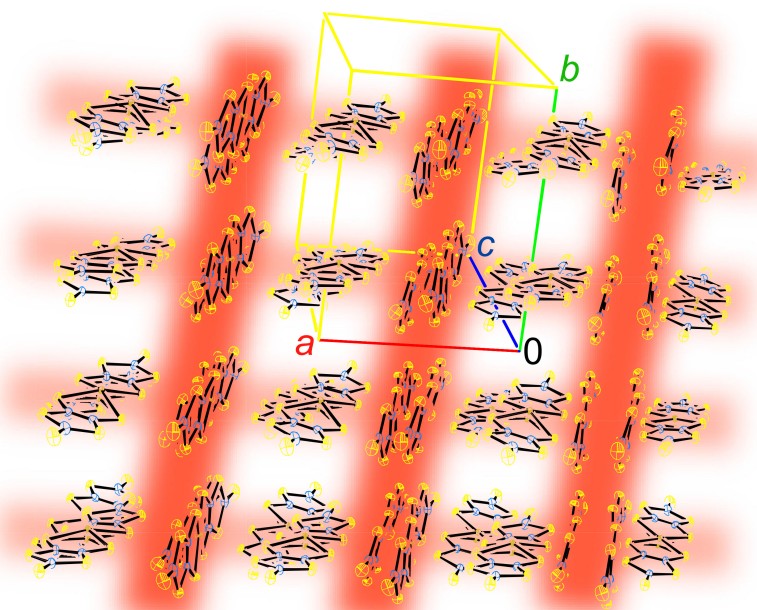

**Figure 11.** Interaction network between the Au(dmit)$_2$ anions in BPY[Au(dmit)$_2$]$_2$. The yellow, blue, and light brown spheres designate sulfur, carbon, and gold atoms, respectively. BPY cations are omitted for clarity. Reproduced from Ref. [13] with permission from the Royal Society of Chemistry.



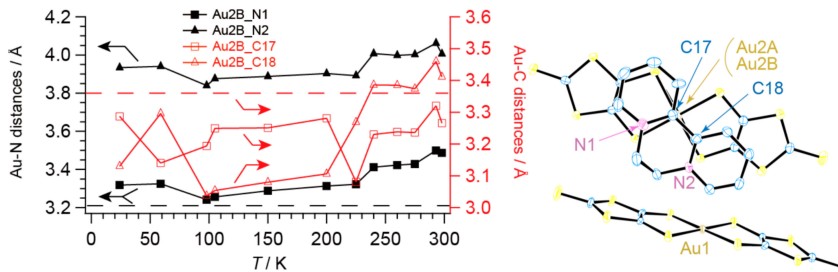

**Figure 12.** Interatomic distances between Au2B and N or C atoms on BPY. Horizontal broken lines indicate the van der Waals distances of Au–N (black; 3.21 Å) and Au–C (red; 3.36 Å), respectively. Each symbol includes uncertainty (esd) of the distance, which is in the order of 0.01 Å. Reproduced from Ref. [13] with permission from the Royal Society of Chemistry.

Hence, we conclude that interaction exists between $Au(dmit)_2$ and BPY. Consistent with these observations, extended Hückel tight-binding band calculations show that **1** is a band insulator ($\delta = 1$ in $[Au(dmit)_2]^{\delta-}$) in the high-temperature (HT) phase ($T > 280$ K), while it contains a small number of unpaired electrons ($\delta \approx 0.99$ in $[Au(dmit)_2]^{\delta-}$) in the low-temperature (LT) phase ($T < 280$ K) [13]. This calculation is also supported by the magnetic susceptibility and the electron spin resonance [13] (see below).

## 2.2. Charge, Spin, Coordination Geometry, and Cation-Anion Interaction

In order to clarify the differences in electronic states between P and NP geometries, we carried out a quantum chemistry calculation based on the molecular structure observed at 100 K (Figure 13) [13]. The results show that there are not so many differences between the two geometries in the spin distribution and molecular orbitals. The energy difference between the two geometries (NP is higher by 1 eV) is smaller than that between the two charges (neutral state is higher by 4 eV) [13]. Meanwhile, because of the shorter distance between the Au atom in $[Au(dmit)_2]^-$ and one of the C atoms in $BPY^{2+}$, the NP geometry favors CT interaction between $[Au(dmit)_2]^-$ and $BPY^{2+}$, which lowers the energy of the electronic system. Such proximity in energy of the two geometries results in an equilibrium between them in the crystal, which enables a flexible and fine tuning of total energy by varying the geometry. Here the equilibrium coexistence of the two geometries is important because their ratio will vary depending on the thermodynamic conditions and in response to perturbations.

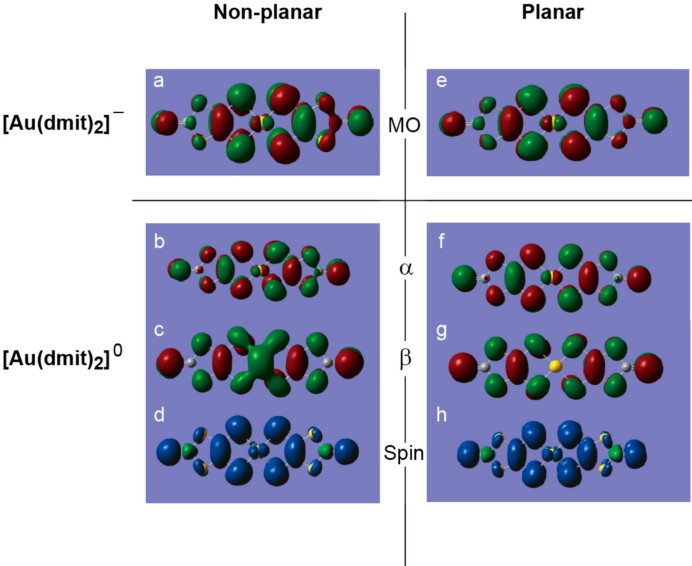

**Figure 13.** DFT (density functional theory) calculations (B3LYP/LanL2DZ(5d,7f)) on P and NP $[Au(dmit)_2]^{n-}$ ($n = 0, 1$) complexes based on the molecular structure determined at 100 K. Here Au2A

DFT (density functional theory) calculations (B3LYP/LanL2DZ(5d,7f)) on P and NP $[Au(dmit)_2]^{n-}$ ($n = 0, 1$) complexes based on the molecular structure determined at 100 K. Here Au2A and Au2B refer to the P and NP Au atoms in the complex anions that face the BPY cations in the crystal, respectively. (**a**) HOMO of $[Au2B(dmit)_2]^-$. (**b**) $\alpha$-Spin SOMO of $[Au2B(dmit)_2]^0$. (**c**) $\beta$-Spin SOMO of $[Au2B(dmit)_2]^0$. (**d**) Electron density from spin SCF density in $[Au2B(dmit)_2]^0$. (**e**) HOMO of $[Au2A(dmit)_2]^-$. (**f**) $\alpha$-Spin SOMO of $[Au2A(dmit)_2]^0$. (**g**) $\beta$-Spin SOMO of $[Au2A(dmit)_2]^0$. (**h**) Electron density from spin SCF density in $[Au2A(dmit)_2]^0$. In Figure 13a–g red and green lobes designate the opposite atomic charge distributions to each other (Mulliken charge; red = −0.753, green = +0.753), while yellow and grey (parts of) spheres are (parts of) gold/sulphur and carbon atoms without significant electron densities. Similarly in Figure 13d,h, the blue and green lobes designate the opposite electron densities to each other (blue = positive, green = negative). In Figure 13, note that the HOMO is based on the monoanion's electron configuration, and thus corresponds to the LUMO of the neutral complex molecule. Since HOMO and LUMO are named/defined on the basis of the neutral molecules in the field of molecular conductors and magnets, care must be taken when readers compare the figure above with those in other references. Reproduced from Ref. [13] and modified with permission from the Royal Society of Chemistry.

Such an active role or involvement of the metal centers in the $\pi$-conjugated partially occupied orbitals is characteristic to the $[Au(dmit)_2]^{n-}$ ($0 \leq n < 1$) complexes, making a stark contrast with other metal-dithiolene complexes [25,65].

## 2.3. Phase Transition

Owing to the many degrees of freedom associated with each other (Section 2.2), it is the phase transition that **1** distinguishes itself from other materials ever reported. Here we shall overview the unique aspects of the phase transition observed in different physical properties connected with different degrees of freedom. Regarding the degree of freedom in molecular structure, on cooling under the dark, the Au2B site occupancy (Occ) suddenly decreases from ~14% ($T \geq 280$ K) to ~8% ($T \leq 200$ K) [13]. Interestingly, the Occ exhibited the opposite behavior in the dark and under UV-irradiated conditions [13]. The observed structural behavior is surprising for the following three reasons.

Firstly, ordinary incoherent UV light is able to realize coherent structural modulation as an extremely long-lived photoexcited state. The lifetimes of photoexcited states are typically an order of $10^{-15}$ to $10^{-9}$ s, while the X-ray structural analysis required a few seconds per one shot of oscillation photograph in the experiments. During taking the photographs, all the atoms in the single crystal are supposed to be static except for thermal displacement around the equilibrium positions. In addition, the X-ray structural analyses were carried out by taking several hundred oscillation photographs for a few seconds each and gave consistent and reproducible results for more than 33 single crystals analyzed. However, the photoexcitation by UV light without coherence may cause random displacement of atoms leading to the breakdown of three-dimensional ordering of atom positions required for the X-ray diffraction study. In addition, the anisotropy in light absorption as well as the low symmetry in morphology of the crystals make the irradiation effects inhomogeneous in the crystals, leading each single crystal to an incoherent assemble of domains being far from a single crystal. As a result, X-ray structural analyses of the crystals under incoherent irradiation often fail to unravel the photoexcited structures. Accordingly, the successful structural analyses suggest that the structural change during the photoexcitation should occur as if the displacement of gold atoms in $Au(dmit)_2$ in the crystals synchronizes with each other with three-dimensional order. This may be connected with the extremely long relaxation times from the UV-excited states. If the relaxation times should be of ~$10^{-15}$–$10^{-9}$ s, then the probability of structural long-range ordering occurring under irradiation by a standard Hg/Xe lamp would be practically negligible, and diffraction based on the UV-excited structure would not have been observed.

Secondly, a thermodynamic transition at a similar temperature to that observed in the dark occurs under continuous UV irradiation. However, the phase behavior is totally vice versa compared with that under the dark condition (Figure 14) [13]. The UV irradiation promotes the system to a highly excited state that is thermally inaccessible. Considering the energy scales involved, namely light (250–450 nm), at ~32,000–58,000 K vs. $T_C$ (280 K), it is surprising that such a small energy difference between the phases can affect the state of matter as well as it does in the dark. Except for Occ, the crystal structures of **1** in the HT and LT phases are almost identical. This type of transition often involves a change in structure, such as molecular arrangement, crystal symmetry, and cell constants, which is not the case here. This is also a surprising aspect of the phase transition.

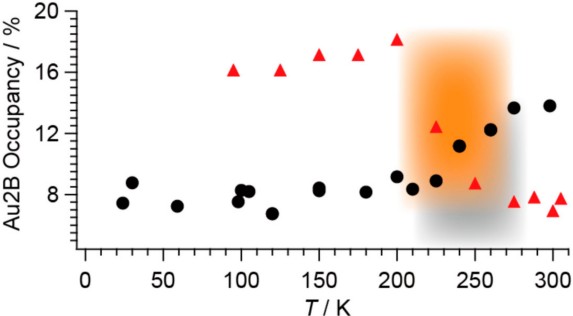

**Figure 14.** Temperature-dependence of the Au2B site occupancy (Occ (%)) under dark (black circles) and UV-irradiated (red triangles) conditions. There are two of crystallographically independent Au sites in BPY[Au(dmit)$_2$]$_2$, Au1 and Au2. Together, Au2A (P) and Au2B (NP) form Au(dmit)$_2$ anions that face the BPY cations. The remaining Au atoms (Au1) are not disordered. Grey and orange square shades show the transition temperature ranges under dark and UV irradiated conditions, respectively. Reproduced from Ref. [13] and modified with permission from the Royal Society of Chemistry.

Thirdly, the phase transition is also unique from the viewpoint of Ehrenfest classification. In general substances, the structural transitions are of the first order, where the volumes suddenly change at the transition temperatures in a discontinuous manner. However, **1** does not exhibit any discontinuous change in the unit cell volume $V$ at ~280 K whether under the dark or UV-irradiation, as $V$ is independent of Occ (Figure 15).

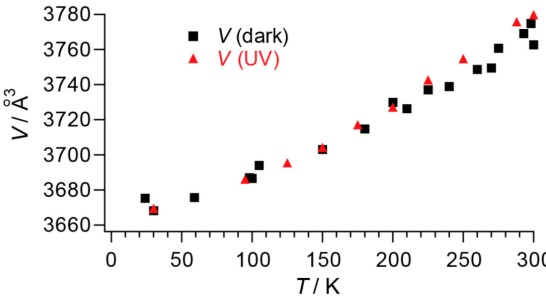

**Figure 15.** Temperature-dependence of cell volume of BPY[Au(dmit)$_2$]$_2$. Squares (black) and triangles (red) are data obtained in the dark and under UV irradiation, respectively.

Such structural uniqueness is closely associated with charge and spin degrees of freedom. Based on the magnetic susceptibility and crystal structure data, the phase transition is characterized by a change in the amount of CT ($q$) between [Au(dmit)$_2$]$^{(1-q)-}$ and BPY$^{2(1-q)+}$; $q \sim 0 \leftrightarrow$ (HT) $q = 0.01$ (LT) and diamagnetic (HT) $\leftrightarrow$ paramagnetic (LT) (Figure 16) [13]. In such an electronic phase transition (between closed-shell and open-shell molecular species) the charge and spin usually lose their degrees of freedom in the LT phase, which is the opposite behavior to that observed in the present case. With regard to charge and spin degrees of freedom, the electronic system in **1** behaves like a liquid that freezes when warmed.

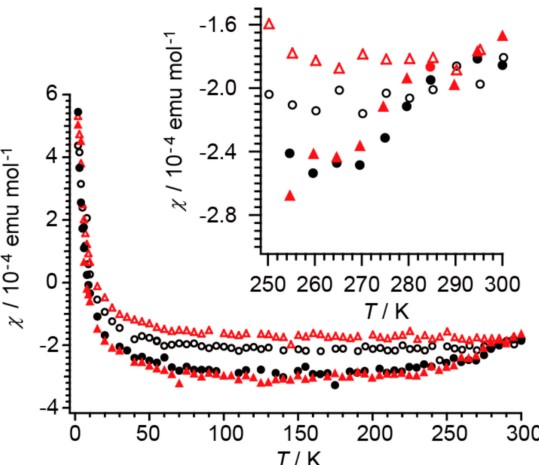

**Figure 16.** Temperature-dependence of magnetic susceptibility. Circles (black) and triangles (red) are data obtained in the dark and under UV irradiation, respectively. Open and closed symbols correspond to ZFC and FC data, respectively. The data include diamagnetic contributions from the core electrons of BPY[Au(dmit)$_2$]$_2$ (inset: enlarged view at $T_C$ ~280 K). Reproduced from Ref. [13] with permission from the Royal Society of Chemistry.

Regarding interesting magnetic transitions, Au(tmdt)$_2$ also exhibits an AF transition at 100–110 K retaining high conductivity down to LT [62–64]. This observation suggests that Au(tmdt)$_2$ should possess both carriers and localized spins in the ground state. Such situation is possible only when the interaction between the metal center (Au) and $\pi$-ligands (tmdt) is appropriately strong/weak. This is based on the close proximity of their energy levels of the atomic/molecular orbitals involved. The Au atoms are the metal of choice for such an electronic structure of metal-complexes with a variety of dithiolene ligands. We can expect a further variety of novel solid-state properties by exploring the AuL$_2$ derivatives with new types of dithiolene ligands.

## 2.4. Unprecedentedly Long Relaxation Time of the UV-Excited State

Now we have reviewed that **1** has unique physical properties based on the close connection between spin, charge, and structural degrees of freedom. As the most prominent feature of the interconnected physical properties, **1** has been found to exhibit an unusually long relaxation time upon UV irradiation, which amounts to a time longer than typical relaxation times in luminescence by more than eight orders of magnitude [1]. Here, one should note that the actual relaxation may consist of a series of different relaxation steps and that some of them may be as fast as ever known. Yet, here, we shall simply call the total relaxation process(es) "relaxation" unless it is misleading. Owing to such extremely slow relaxation, it was conceived that one could not apply various established methods of time-resolved ultrafast spectroscopy to analyze this phenomenon, which generally cannot resolve the phenomena slower than tens of nanoseconds or more [78–94]. Thus, we measured electron spin resonance (ESR) spectra to estimate the number of residual UV-excited unpaired electrons after cessation of UV-radiation.

After UV-irradiation for 30 min at 298 K, the UV lamp was turned off. Then, the ESR spectra under the dark condition were repeatedly measured for the same single crystal at 298 K. Along with the ESR spectra under UV irradiation around $T_C$ (Figure 17a,b) [13], the result is shown in Figure 17c as the ESR intensity vs. elapsed time (h) [13]. From the curvature of the decay curve, it is apparent that there should be at least two different relaxation times, $\tau_i$ ($i$ = 1, 2).

$$\tau_1 = 2.98 \pm 0.68 \ (h)$$
$$\tau_2 = 36.4 \pm 17.7 \ (h)$$

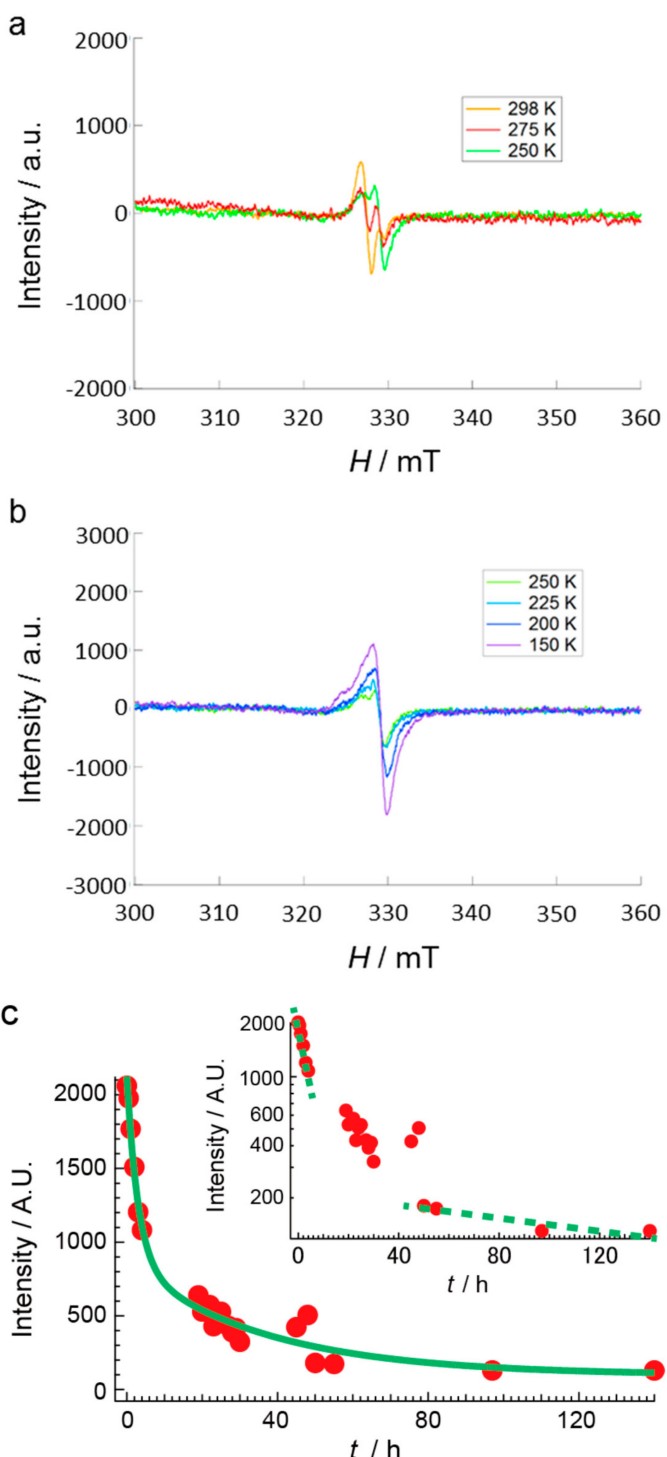

**Figure 17.** The X-band (9.3 GHz) electron spin resonance (ESR) measured using the single crystal of **1**. The magnetic field was applied in the direction of the *a*-axis of the single crystal, and the UV was irradiated on the *ab*-plane of the single crystal. Temperature-dependencies of the spectra under UV-irradiation at (**a**) 298 K, 275 K, and 250 K, and at (**b**) 250 K, 200 K, and 150 K. The spectrum at 250 K appears in different panels for comparison of signal intensity. (**c**) Time-dependence of electron spin resonance (ESR) signal intensity after cessation of UV-irradiation at 298 K. (Main panel) Intensity vs. Time (*t*/h) plot. Red circles and the green curve designate observed and fitting data using a two-component exponential decay model, respectively. (Inset) log(Intensity) vs. Time (*t*/h) plot. Red circles and two green broken lines designate observed and fitting data using a two-component exponential decay model, respectively. The inset clearly shows that there are fast and slow processes in the observed relaxation. Reproduced from Ref. [13] with permission from the Royal Society of Chemistry.

It should be noted that the relaxation times $\tau_i$ are different from usual meaning in ESR spectroscopy ($\tau_{ESR}$); the latter ($\tau_{ESR}$) is associated with relaxation from the microwave-excited spin state, while the former ($\tau_i$) is associated with relaxation from the UV-excited state.

Although the errors in the determined values of $\tau_i$ are large, both relaxation times are unusually long for a photoexcitation process. Even at 298 K, it took several days (~140 h) to recover the original intensity before irradiation, during which time the energy received from the photons was stored in **1** and gradually released as heat, suggesting a complicated mechanism involving structural changes. Meanwhile, the molecular and crystal structures remain unchanged under UV-radiation, with the exception of an obvious change in Occ upon UV-irradiation, from ~8% (dark) to ~14% (UV) at 300 K, for example (Figure 14) [13]. Considering the general scheme of photoluminescence (Figure 1) [4] as well as the results of the X-ray structural analyses (Figures 12 and 14) [13], the magnetic susceptibility (Figure 16) [13], and the ESR (Figure 17) [13], an excitation from $S_0$ to $S_1$ and successive ISC from $S_1$ to $T_1$ are considered to occur in **1** under UV irradiation. However, **1** does not emit light after UV irradiation. Therefore, there should be some non-radiative decays involving a series of trapping states via tunneling and/or thermal processes similar to the PersL (Figure 2) [5]. However, the band gap (~1.2 eV) between the valence (HOMO) and conduction (LUMO) bands in **1** (under the dark condition) is clearly less than the UV-vis photon energies (~1.6–5 eV), and the band widths are almost negligible (~0.01 eV). These features make it difficult to apply the PersL mechanism (Figure 2) directly to **1**. On the other hand, if we should interpret the observed relaxation in **1** (Figure 17c) [13] on the basis of the LPL scheme (Figure 3) [11], the extremely slow decay process should contain repetitive exchange between excited states and intermediate states. In fact, the intermediate states are considered to include a partially charge separated state formed by a pair of radical cation and anion dissociated from photo-generated exciplex [11]. It is not surprising if a similar situation should take place in the UV-excited states in **1**. However, the LPL scheme requires a power-law decay, while a standard phosphorescence exhibits an exponential decay. The observed decay (Figure 17c), though it does not involve an emission process, is reproduced by an exponential function [13]. Thus, the relaxation mechanism of **1** could be different from any of known or proposed mechanisms. Evidently, further study is required for understanding the excitation and relaxation mechanisms of **1** upon UV irradiation.

Salt **1** is expected to enable energy storage by molecular distortion, which will be realized when we find how to completely control the distortion. Relaxation generally becomes slower with decreasing temperature. In addition, the phase transition may play an important role in energy storage, where the stability reverses between P and NP geometries. The next goal involves discovering related materials that have even longer relaxation times in addition to finding a trigger to make them emit light on demand.

## 3. Comparative Study on Related Materials

### 3.1. Selection of the Counter Cations

In this section, we shall discuss our recent attempt [95] to obtain the related salts to **1**. By comparing them with **1**, we will discover appropriate cations and key factors for the materials for PES. As discussed thus far, the CT interaction between the cations and [Au(dmit)$_2$]$^-$ anions is indispensable for producing NP coordinate Au atoms in the Au(dmit)$_2$ anion salts. The selected cations (Figure 18) have closely related molecular structures to that of BPY$^{2+}$, still systematically differ from each other in such aspects as electric charge, symmetry, structural flexibility, and redox reactivity under the dark and/or under UV irradiation. Some of them have also similar molecular structures to biphenyl, which is well known as a photoluminescent compound under UV radiation [96–98]. First, we carried out single crystal X-ray structural analysis of all the obtained salts both under the dark and under the UV-irradiated conditions. Selected structures are shown in Figures 19 and 20. Then we examined the intermolecular interactions between the cations and anions by calculation [95].

**Figure 18.** Cations examined in the syntheses of the [Au(dmit)$_2$]$^-$ salts. MnPp$^+$ = *N*-methyl *n*-phenylpyridinium (*n* = 2–4), NMQ$^+$ = *N*-methyl quinolinium, NMP$^+$ = *N*-methyl phenazinium, Phen$^{2+}$ = *N,N'*-ethylene 1,10-phenanthrolinium, M34B$^+$ = *N*-methyl 3,4-benzoquinolinium, M56B$^+$ = *N*-methyl 5,6-benzoquinolinium, M78B$^+$ = *N*-methyl 7,8-benzoquinolinium.

### 3.2. Comparison of the Resultant Salts

Now we will discuss possible key factors for the photoresponsive NP-Au(dmit)$_2$ anions in the crystalline CT salts. The difference in molecular geometries (isomers) and the flexibility of the molecular structures (co-planarity) did not result in systematic variation in the cation-anion CT interactions (Figures 19 and 20). Since the energy scale of the transfer integrals in this kind of molecular crystal (~0.1 eV) is clearly different from that of UV photons (~3–5 eV), the interactions between the frontier and higher energy orbitals could be important under UV excitation. Thus, we paid attention to the transfer integrals, $t_{12}$, between the two MOs, MO1 and MO2, with the energy separations in the range of the UV excitation (~250–450 nm ≈ 2.76–4.96 eV). Evidently, for both P- and NP-Au(dmit)$_2$ anions under both dark and UV-irradiated conditions, **1** possesses a far larger number of significant values of $t_{12}$ than the MXB salts (X = 34, 56, and 78) [95]. Although most of the corresponding optical transitions are forbidden or weak, such differences should lead to substantial differences in the amount of CT between the cations and anions under UV irradiation due to higher order of perturbations, which could cause a qualitative difference in the occurrence of the NP-Au(dmit)$_2$ between the MXB and BPY salts.

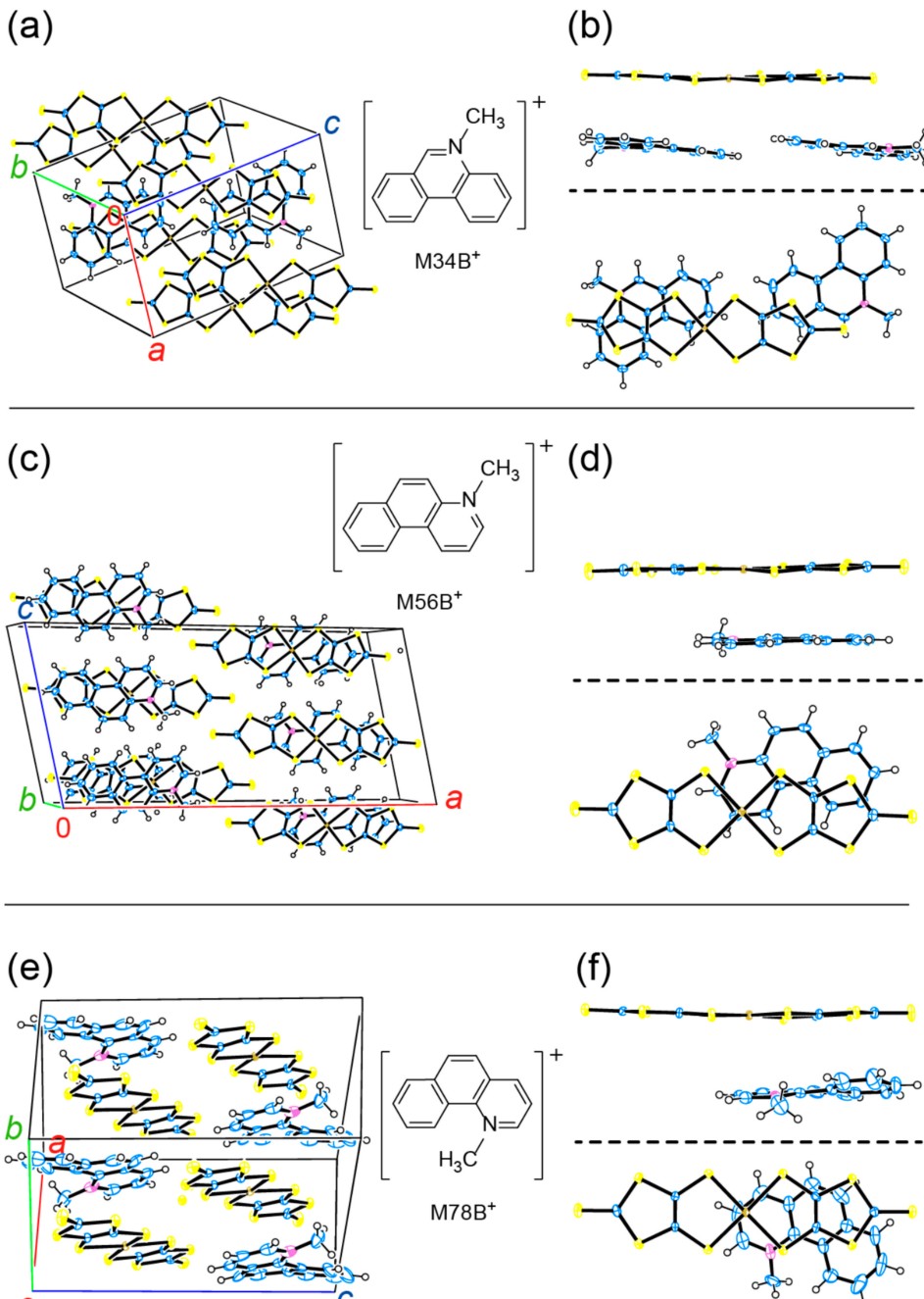

**Figure 19.** Typical molecular arrangements of anions and cations in MXB[Au(dmit)$_2$]. For MXB$^+$, see Figure 18. (**a,b**) X = 34; (**c,d**) X = 56; (**e,f**) X = 78. Blue, purple, blank, yellow, and brown spheres designate C, N, H, S, and Au atoms, respectively. Reproduced and modified from Ref. [95] with permission from the Chemical Society of Japan.

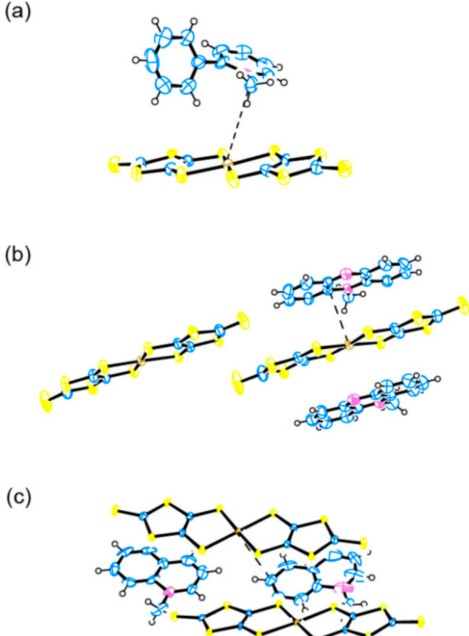

**Figure 20.** Typical molecular arrangements of anions and cations in (**a**) M2Pp[Au(dmit)$_2$], (**b**) NMP[Au(dmit)$_2$], and (**c**) NMQ[Au(dmit)$_2$]. For M2Pp$^+$, NMP$^+$, and NMQ$^+$, see Figure 18. Blue, purple, blank, yellow, and brown spheres designate C, N, H, S, and Au atoms, respectively. The broken lines indicate the Au–C$_N$ distances; (**a**) 4.40(1), (**b**) 3.584(4), and (**c**) 3.726(6) Å. The dihedral angle between the two aromatic rings in the M2Pp$^+$ cation in (**a**) is 58.6(16)°, while the NMP$^+$ and NMQ$^+$ cations and all the [Au(dmit)$_2$]$^-$ anions are almost perfectly planar. Details are deposited to Cambridge Crystallographic Data Centre; the CCDC # are (**a**) 2032445, (**b**) 2032448, and (**c**) 2032449.

Another important factor is the charge distribution on each molecule of the interacting pair of cations and anions. As the atoms in the facing cations nearest from the Au atoms are carbon atoms in all the salts examined, we shall call them C$_N$ atoms below. The DFT (density functional theory) calculations showed that the charges on the C$_N$ atoms clearly differ from each other between the BPY and MXB salts [95]. In the BPY salt, the charge on the Au atom and that of the C$_N$ atom had opposite signs to each other, while they had the same sign with each other in the MXB salts. While the opposite charges between two atoms favor the coordination bond formation, Coulombic repulsion exists between the atoms having charges of the same sign, hindering such bond formation, especially when the bond is weak as in the Au(dmit)$_2$ salts. The charges on the Au (~−0.8 e$^-$) and C$_N$ atoms (~−0.3 e$^-$) were nearly independent of cations except for the BPY cation (~+0.4 e$^-$). In **1**, the C$_N$ atom was located at the position enabling sufficient overlap between C 2p and Au 6p orbitals. In the MXB salts, none of the C atoms in the cations satisfied the requirement of the location and the sign of charge for the C$_N$ atoms. Based on the comparison thus far, the promising candidate cations should have the following features. (i) They are to have large positive charges on the C$_N$ atoms as well as to form 1:2 salts. (ii) They do not have inversion centers for making two [Au(dmit)$_2$]$^-$ species crystallographically independent in the crystals to realize similar molecular packing patterns to that in **1** (Figure 9). (iii) They have positively charged atoms like the C$_N$ atoms around the centers of the molecules to form coordination bonds with the Au atoms in [Au(dmit)$_2$]$^-$ species. (iv) Regarding the amount of the positive charge on the C$_N$ atoms, the larger, the better, because the larger charge favors the coordination bond formation between the negatively charged Au and the positively charged C$_N$ atoms. By calculating the charge distributions on the cations, one may narrow down the appropriate cations for the photoresponsive NP-Au(dmit)$_2$ anions by paying attention to the possible C$_N$ atoms. Consistently, the DFT calculation based on the structures under the dark condition showed that cation-anion interaction stabilizes the [NP-Au(dmit)$_2$–BPY]$^+$ pair by 0.50 eV compared to the [P-Au(dmit)$_2$–BPY]$^+$ pair, but that the isolated

[NP-Au(dmit)$_2$]$^{n-}$ species are less stable by 1.0 eV than the isolated [P-Au(dmit)$_2$]$^{n-}$ for both $n = 0$ and 1. Approximately, this means that the weak Au–C bond stabilizes the [NP-Au(dmit)$_2$]$^{n-}$ species by 1.5 eV, i.e., ~1/2–1/3 of a covalent bond energy.

Now we shall prove the validity of this energy by showing that the enthalpy change between the high-temperature and low-temperature phases approximately accounts for the corresponding entropy change. Since this is not a first-order phase transition, there is no enthalpy of transition (latent heat), i.e., a discontinuous enthalpy change at the transition temperature $T_C$. Even in such a case, one can similarly estimate the differences in enthalpy $\Delta H$ and entropy $\Delta S$ between the high-temperature and low-temperature phases, as they are state functions, being independent of paths such as transitions connecting the two states. If we assume that the weak bond energy estimated above (~1.5 eV) should be associated with $\Delta H$ and that $\Delta\text{Occ} = 14 - 8 = 6\%$ of the [Au(dmit)$_2$] molecules in a crystal containing 1 mole of [Au(dmit)$_2$] molecules should form/dissociate the weak bond during the transition under the dark condition,

$$|\Delta H| = 1.5\,[\text{eV}] \times 96.485\left[\text{kJ mol}^{-1}\text{eV}^{-1}\right] \times 0.06 \cong 8.68\left[\text{kJ mol}^{-1}\right] \tag{1}$$

Thus, the corresponding entropy change $\Delta S$ is calculated as follows by noting that the phase transition proceeds at a constant pressure as a reversible change.

$$|\Delta S| = \int_{T_1}^{T_2} \frac{dH}{T} \cong \frac{1}{T_C}\int_{T_1}^{T_2} dH = \frac{\Delta H}{T_C} \approx \frac{8.68 \times 10^3}{250} \cong 35\left[\text{JK}^{-1}\text{mol}^{-1}\right] \tag{2}$$

where $T_1$ and $T_2$ are the initial and final temperatures of the transition, and $T_C \sim 250$ K is the mid-point estimated based on the occupancy (Occ (%)) in Figure 14 [13].

On the other hand, $\Delta S$ can be also calculated as the statistical entropy as follows:

$$|\Delta S| = \frac{1}{2}N_A k_B ln\frac{W_f}{W_i} \tag{3}$$

$$W_i = {}_{100}C_8 = \frac{100!}{92!8!}, \; W_f = {}_{100}C_{14} = \frac{100!}{86!14!} \tag{4}$$

where $N_A$, $k_B$, $W_i$, and $W_f$ are Avogadro's constant, Boltzmann's constant, and the initial and final numbers of microstates, respectively. The coefficient 1/2 in Equation (3) is required because only half of the Au atoms (Au2 only) in the [Au(dmit)$_2$] molecules in the crystal can be disordered. By substituting the appropriate values in Equation (3),

$$|\Delta S| = \frac{1}{2}N_A k_B \ln\frac{W_f}{W_i} \cong 51\left[\text{JK}^{-1}\text{mol}^{-1}\right] \tag{5}$$

Fair agreement between Equations (2) and (5) suggests that the calculated energy levels should be consistent with each other. The same calculation semi-quantitatively accounts for the slight lowering of the observed $T_C$ under UV irradiation (Figure 14) [13], where the Occ changes between ~17% and 8%.

In summary, the systematic and comparative study above revealed that the characteristically strong CT interactions between BPY$^{2+}$ and [Au(dmit)$_2$]$^-$ species are based not only on the frontier orbitals but also on various other MOs to be (partially) occupied by the photoexcited electrons. In addition, the DFT calculation has elucidated that such strong CT interactions originate from the charge distribution characteristic to BPY$^{2+}$, leading to the cation-anion overlapping mode advantageous for the occurrence of the NP-Au(dmit)$_2$ anions and the Au–C bond formation as well as the consequent unique long-lived photoexcitation.

## 4. Future Prospects

Science is often driven by a long-term goal that lies between reality and dream. Although it may be too early to specify examples of possible applications of the study discussed thus far, the author would like to discuss how the current study could eventually change the quality of human lives.

The merits of the photon storage materials will be immeasurable if people can store and carry photons in a material and use them when and where they wish. However, this is evidently difficult. Although the number of functions offering convenience performed by wireless devices such as automatic doors and cellular phones has increased, and thus the number of electrons substituted by photons in the semiconductor technology has increased, most of the optical devices still require electric devices to perform their tasks. For example, solar panels can generate electricity from sunlight, but they cannot store electricity or photon energy within themselves. Thus, they require other devices to use or store the generated electricity. Why can we not store photons?

If a material can store photons, it should store the energy of photons in them without releasing it as heat or light until a trigger is provided to the material to emit light. Such a material must possess two incompatible properties: it should be an inferior photoluminescent material, because it should not release energy by emitting light until a trigger is provided. Additionally, it should be a superior photoluminescent material because it should immediately emit light when necessary. We assume the phase transition in **1** to be a key feature for reconciling the two requirements regarding the photoluminescent properties above, as the energy balance between P and NP geometries in the molecular crystal completely reverses across the transition. Most photoluminescent materials involve some distortions in molecular structure in the photoexcited states, and the emission occurs during or as a result of the recovery of the original structures by releasing the distortion energies. If one can control the emission, i.e., one can not only interrupt and resume the emission, but also increase and decrease the intensity of emission as they wish, the photons can be used in a similar way as electricity (electrons) generated from batteries. Most of the known photoluminescent materials spontaneously emit light of different (generally longer) wavelengths from those they received, owing to the energy gain or loss by photon–phonon interactions. Few exceptional luminescent materials emit the same wavelengths of light with those they received, e.g., resonance fluorescence in Hg vapor (185 nm). Nonetheless, at present, the duration as well as the commencement and cessation of emission cannot be controlled once the material has received the appropriate light wavelength. These features prevent us from replacing electric currents (electrons) in integrated circuits for computers by light beams (photons). To realize more advanced information technologies with less electric power consumption and with a smaller and lighter device, using photons in place of electrons may be one of the solutions, which consequently requires a novel material with controllable emission properties. This is known as "photon storage material" herein. If the wavelengths are not changed by the storage, they would be favorable for the wireless transportation of energy and information at a distance through a series of devices. If such a material is discovered, many technological advances can be realized. For example, in computers, the stored photons can be used as a power source in place of batteries as well as to perform variable tasks simultaneously. It is noteworthy that a single photon of UV range, for example, contains a larger amount of energy than other types of energy resources such as batteries. If necessary, photons can be transformed into heat by a hundred percent, while it is impossible to transform heat into light by the same percentage (forbidden by the second law of thermodynamics). The most prominent usage is in wireless devices and the quickest transportation of energy, which distinguishes light from other life necessities such as gas, water, and electricity. Hence, photon storage materials, for example, will serve as a temporal power and light source when electricity is not available during an emergency.

We are currently studying a wider range of related materials. Based on the findings of **1**, the next step is to reveal the mechanism of relaxation after UV-radiation. We will discover new materials that exhibit longer relaxation times by physical (thermodynamic condition control) or chemical (synthesis) approaches. Additionally, we are investigating the method to control the emission after storage, i.e., the trigger for emission. The results will be reported in due course.

**Funding:** This work was partially supported by Grant-in-Aid for Challenging Exploratory Research (18K19061) of JSPS, the Tokyo Chemical Industry Foundation, the Tokyo Ohka Foundation for the Promotion of Science and Technology, the Kato Foundation for Promotion of Science, and an Ehime University Grant for Project for the Promotion of Industry/University Cooperation.

**Acknowledgments:** The assistance of Kensuke Konishi (SQUID), Keishi Ohara (ESR), Shigeki Mori and Rimi Konishi (X-ray structural analyses) at Ehime University is acknowledged.

**Conflicts of Interest:** The author declares no conflict of interest.

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
