# Peer review of "Prototype Material for New Strategy of Photon Energy Storage"

_inorganics, doi:10.3390/inorganics8100053_

Round 1

Reviewer 1 Report

This is an excellent account on prototype materials for photon energy storage. After a review on anionic and neutral gold bis(dithiolene complexes, the authors discuss comprehensively their own results on anionic Au(dmit)2 complexes with a special focus on the one with the BPY dication. Clearly there is still a long path until the discovery of efficient organic materials acting as photon reservoirs, yet the authors give here an interesting personal insight into this field. I recommend acceptance of the manuscript.

Author Response

Thank you very much for your positive and high evaluation of our manuscript.

Reviewer 2 Report

This paper by Naito describes the author’s main advances in the search for photon energy storage (PES) materials, based however on a non-emissive system investigated by the author for several years, namely the [bpy2+][Au(dmit)2-]2.  

The topics is timely and fits well in the subjects covered by Inorganics.

A first part (1/3 of total length) is dedicated to extensive bibliographic investigations related to (i) PES, and (ii) gold dithiolene complexes (including 73 references), a second part (1/3 of total length) is dedicated to already reported results (mainly in ref 14) concerning [bpy2+][Au(dmit)2-]2, a third part  (1/3 of total length), is dedicated to already reported (mainly in Ref 98) alternative systems where the [bpy2+] dication is replaced by other systems.

Altogether, this paper is interesting and worth publishing as it gathers together the enthusiastic ideas and recent results of Pr. Naito’s research but it should be considered as a Review and NOT an Article or Feature Article as in includes essentially no new results at all.

Otherwise a few typing errors I identified:

Page 1l, line 274-275: Their single crystals ….and being stable …

Page 22, line 547: structural

Page 22 line 552: structures

Page 22, line 558: Thus we examined in more details the intermolecular …

Page 26, line 633: significant

Author Response

(Reviewer 2; Comment 1)

This paper by Naito describes the author’s main advances in the search for photon energy storage (PES) materials, based however on a non-emissive system investigated by the author for several years, namely the [bpy2+][Au(dmit)2-]2.  

The topics is timely and fits well in the subjects covered by Inorganics.

A first part (1/3 of total length) is dedicated to extensive bibliographic investigations related to (i) PES, and (ii) gold dithiolene complexes (including 73 references), a second part (1/3 of total length) is dedicated to already reported results (mainly in ref 14) concerning [bpy2+][Au(dmit)2-]2, a third part (1/3 of total length), is dedicated to already reported (mainly in Ref 98) alternative systems where the [bpy2+] dication is replaced by other systems.

Altogether, this paper is interesting and worth publishing as it gathers together the enthusiastic ideas and recent results of Pr. Naito’s research but it should be considered as a Review and NOT an Article or Feature Article as in includes essentially no new results at all.

(Reply to Comment 1)

Thank you very much for your positive and high evaluation of our manuscript. As pointed out, this manuscript is meant for a review, not an original paper or article, of course. As I was first invited by one of the Guest Editors to submit a Feature Article, and then I replied and promised to submit a review paper instead of an article. The contradiction between the type of the manuscript and the actual content happened in this way.

(Reviewer 2; Comment 2)

Otherwise a few typing errors I identified:

Page 1l, line 274-275: Their single crystals ….and being stable …

Page 22, line 547: structural

Page 22 line 552: structures

Page 22, line 558: Thus we examined in more details the intermolecular …

Page 26, line 633: significant

(Reply to Comment 2)

Thank you very much for your kind and careful review. I have thoroughly checked the manuscript and all the five typos above in addition to the following errors have been corrected, which are highlighted by red letters with the track-change in the revised manuscript.

Page 18, L. 444 transition

(The line and page numbers are those in “inorganics-925122_Rev1_1”)

Reviewer 3 Report

The current manuscript by Naito presents a detailed account of the properties and comparative study of BPY[Au(dmit)2]2. It presents the structural analysis performed on the crystal of the salt, and suggests that the structural disorder might the reason why the structure could have a long-lived excited state with very slow relaxation. It also presents a number of other analyses, followed by comparative study on a number of other crystals with related structures.

The interesting studies aside, this manuscript as a "review" does not provides too much new insight or perspective other than the two papers the author rely heavily on (Ref [14] and [98]). Most of the details are reproduced from the two published papers by the author, without minimal high-level thinking that should be present in a review-type manuscript. In fact, the citations in Section 2 and 3 are so minimal that one might mistakenly regard that as an original research at the first glance of the manuscript.

I would therefore suggest the author to rearrange the main body of the review, bringing upon a broader picture instead of just copying down the facts from the original manuscript. At the same time, it is also preferable to tone down the introduction. The current introduction has too much hype and speculation by the main subject of the review, BPY[Au(dmit)2]2, is only a small step towards it. The main claim of the introduction, controlled photon emission, was in fact not achievable by the salt under review.

A small point to note on Figure 15 is that there seems to be some problem with panel f, that it seems at least two orbitals were interlaced on the figure (evident upon close examination), that caused one lobe of the orbital on the left hand side (half red and half green) to appear in a wrong phase.

Author Response

(Reviewer 3; Comment 1)

The current manuscript by Naito presents a detailed account of the properties and comparative study of BPY[Au(dmit)2]2. It presents the structural analysis performed on the crystal of the salt, and suggests that the structural disorder might the reason why the structure could have a long-lived excited state with very slow relaxation. It also presents a number of other analyses, followed by comparative study on a number of other crystals with related structures.

The interesting studies aside, this manuscript as a "review" does not provides too much new insight or perspective other than the two papers the author rely heavily on (Ref [14] and [98]). Most of the details are reproduced from the two published papers by the author, without minimal high-level thinking that should be present in a review-type manuscript. In fact, the citations in Section 2 and 3 are so minimal that one might mistakenly regard that as an original research at the first glance of the manuscript.

I would therefore suggest the author to rearrange the main body of the review, bringing upon a broader picture instead of just copying down the facts from the original manuscript. At the same time, it is also preferable to tone down the introduction. The current introduction has too much hype and speculation by the main subject of the review, BPY[Au(dmit)2]2, is only a small step towards it. The main claim of the introduction, controlled photon emission, was in fact not achievable by the salt under review.

(Reply to Comment 1)

Thank you very much for your kind and careful review. I have added discussion and citations in Sections 2 and 3 as much as possible. Although the reviewer stated that this review contains no “high-level thinking”, the theoretical interpretation on the entropy change (LL. 602-624) during the phase transition of BPY[Au(dmit)2]2, for example, has been published in none of the cited references. Such detailed discussion is suitable for a review. Furthermore, the order of phase transition (Ehrenfest classification) is concisely discussed to highlight the uniqueness of BPY[Au(dmit)2]2 (LL. 428-432 with Fig. 15), which was never tried in the previous papers. Similarly, the promising candidates of the counter cations for PES are discussed based on all the data thus far including unpublished results (LL. 586-595), which would not have been done without this review. The decay mechanism is tentatively discussed based on the proposed mechanisms of different types of persistent luminescence introduced in Section 1.1 (LL. 501-518), which has not been published elsewhere. In accordance with these additional discussion, the following figures are newly added ones during this revision or already present in the first version, all of which information have not been published in the cited references.

            Fig. 15 (P. 19), Fig. 18 (P.23), and Fig.20 (P. 25)

(The figure and page numbers are those in “inorganics-925122_Rev1_1”)

 At the same time, I have deleted the details as much as possible to describe the uniqueness of BPY[Au(dmit)2]2 in a concise way and to provide a broader picture. Accordingly the deleted figures and tables are as follows.

              Fig. 12 (P. 15), Fig. 13 (P. 15), and Table 3 (P. 25)

(The figure, table, and page numbers are those in “inorganics-925122_Rev1_1”)

However, I have no choice but specify the necessary details with figures for readers to understand the points clearly without reading the cited references. In order to discuss the decay mechanism from the photoexcited states, which has not been discussed in any of our previous papers and thus is an important topic in this review, the other related relaxation mechanisms should be introduced (Section 1.1). Similarly, the contrast in structural and physical properties between BPY[Au(dmit)2]2 and other related compounds is indispensable in the introduction to explain the unique behaviour of the former (Section 1.2). The introduction is comprised of these two sections and contains no speculation but review of the related studies of each topic except for the final paragraph of nine lines in Section 1.1. About a half (four lines) of the paragraph describes the full name and basic information of BPY[Au(dmit)2]2. The rest of the paragraph briefly explains how the unique properties of BPY[Au(dmit)2]2 might be connected with PES. Although the goal of the research contains speculation at present, this paragraph connects the apparently irrelevant Sections 1.1 and 1.2 as well as the following main subject of the review. Without indicating the goal of the research explicitly in this paragraph (and in Abstract), readers would not understand why the author pays particular attention to unrealized luminescence of BPY[Au(dmit)2]2, which is examined regarding the conducting and magnetic properties. Even if the goal is unachieved and far away from here, the interpretation and discussion of the data are never based on speculation in this review. Thus, in my view, I could not tone down the introduction to answer the request of the present reviewer. Anyway thank you for your kind suggestion.

(Reviewer 3; Comment 2)

A small point to note on Figure 15 is that there seems to be some problem with panel f, that it seems at least two orbitals were interlaced on the figure (evident upon close examination), that caused one lobe of the orbital on the left hand side (half red and half green) to appear in a wrong phase.

(Reply to Comment 2)

Thank you very much for your kind and careful review. I have replaced it with the correct picture.

Round 2

Reviewer 3 Report

I am happy with the author's response.